# An Antigenic Thrift-Based Approach to Influenza Vaccine Design

**DOI:** 10.3390/vaccines9060657

**Published:** 2021-06-16

**Authors:** Jai S. Bolton, Hannah Klim, Judith Wellens, Matthew Edmans, Uri Obolski, Craig P. Thompson

**Affiliations:** 1Peter Medawar Building for Pathogen Research, Oxford OX1 3SY, UK; jai.bolton@keble.ox.ac.uk (J.S.B.); hannah.klim@zoo.ox.ac.uk (H.K.); judith.wellens@ndm.ox.ac.uk (J.W.); matthew.edmans@ndm.ox.ac.uk (M.E.); 2Department of Zoology, University of Oxford, Oxford OX1 3SZ, UK; 3Department of Philosophy, Future of Humanity Institute, University of Oxford, Oxford OX2 0DJ, UK; 4Nuffield Department of Medicine, University of Oxford, Oxford OX3 7BN, UK; 5Translational Research in GastroIntestinal Disorders, University Hospitals Leuven, 3000 Leuven, Belgium; 6School of Public Health, Faculty of Medicine, Tel-Aviv University, Tel-Aviv 69978, Israel; uriobols@tauex.tau.ac.il; 7Porter School of Environment and Earth Sciences, Faculty of Exact Sciences, Tel-Aviv University, Tel-Aviv 69978, Israel

**Keywords:** vaccine, vaccination, influenza, evolutionary theory, antigenic drift, antigenic thrift

## Abstract

The antigenic drift theory states that influenza evolves via the gradual accumulation of mutations, decreasing a host’s immune protection against previous strains. Influenza vaccines are designed accordingly, under the premise of antigenic drift. However, a paradox exists at the centre of influenza research. If influenza evolved primarily through mutation in multiple epitopes, multiple influenza strains should co-circulate. Such a multitude of strains would render influenza vaccines quickly inefficacious. Instead, a single or limited number of strains dominate circulation each influenza season. Unless additional constraints are placed on the evolution of influenza, antigenic drift does not adequately explain these observations. Here, we explore the constraints placed on antigenic drift and a competing theory of influenza evolution – antigenic thrift. In contrast to antigenic drift, antigenic thrift states that immune selection targets epitopes of limited variability, which constrain the variability of the virus. We explain the implications of antigenic drift and antigenic thrift and explore their current and potential uses in the context of influenza vaccine design.

## 1. Introduction

Seasonal influenza causes an estimated three to five million cases of severe disease and between 290,000 to 650,000 deaths per year [1]. Influenza pandemics have also occurred four times since 1918, causing over 50 million deaths [2]. In order to reduce severe illness and death from both seasonal and pandemic influenza, governments and international organisations have expressed their desire to develop a universal influenza vaccine that would protect against all future circulating human influenza strains [3,4]. Here, we review potential routes for the development of a universal influenza vaccine in the context of evolutionary theory. We present two theories describing the antigenic evolution of influenza: the well-known antigenic drift theory, alongside the antigenic thrift theory. These theories and their corollaries can potentially facilitate the development of a universal influenza vaccine.

Influenza A viruses are comprised of eight segments of negative-strand RNA, which encode for up to seventeen proteins. Three of these proteins, haemagglutinin (HA), neuraminidase (NA), and the matrix-2 protein (M2) are situated on the surface of the influenza virion. The most abundant of them is haemagglutinin (HA). HA is the primary determinant of cell entry, binding to the influenza virus receptor. It is estimated to be the target of 60% of anti-influenza antibodies [5]. For these reasons, HA is the target for most seasonal and universal influenza vaccines under development [3,4,5]. NA cleaves sialic acid, allowing the release of virions from infected cells [6], whilst M2 acts as a transmembrane proton channel that changes the internal pH of the virion during infection [7].

The structure of HA can be broadly divided into two domains: the *head* and the *stem*. The head domain is exposed to the immune system and contains the receptor binding site (RBS), which binds to the target cell receptor, sialic acid, initiating endocytosis [8]. Once the virion is engulfed in a vesicle and transported into the target cell, the stem domain initiates fusion of the viral membrane and vesicle [9,10]. As the head domain of the HA is exposed to the immune system, it experiences selective pressure. The stem domain is partially hidden from the immune system by the head domain due to steric hindrance [11]. As a result, the stem is less variable than the head domain [12]. Moreover, several studies used high-throughput PCR-based methods to mutate every residue in the HA structure to every other possible residue. They concluded that the stem domain also has an intrinsically lower capacity to vary than the head domain [13].

There are currently two influenza A subtypes and two influenza B lineages circulating in the human population [14]. Influenza A subtypes are denoted by HA (1–18) and a second surface protein, called neuraminidase (NA, 1–11). New subtypes emerge via antigenic shift, also known as genetic reassortment, by which two different subtypes combine to form a third novel subtype [15,16]. This is most likely to occur in swine, which are susceptible to both avian and human influenza, as it expresses both 2′6 and 2′3 sialic acid, found in humans and birds, respectively [17]. In this instance, if more than one subtype were to infect a cell, it is possible that this process of reassortment may occur during replication [15,17,18]. When antigenic shift occurs, there may be little population immunity to the rearranged subtype [18].

Current seasonal influenza vaccines are comprised of one H1N1 and one H3N2 influenza A strain, as well as one or two strains from the influenza B (IBV) Victoria and Yamagata lineages [19]. These vaccines are produced as split or live attenuated (LAIV) vaccines. Split vaccines are whole viruses which have been inactivated and disrupted by detergents, whilst attenuated vaccines are temperature sensitive, replicating better in the cooler nasopharynx than they do in the warmer lower respiratory tract [20,21,22]. The strains used for the vaccine are determined by the World Health Organization (WHO) around six months prior to the Northern or Southern Hemisphere influenza seasons [23]. Predicting the dominant strain in the following season is challenging and often results in mismatches between the vaccine and the dominant circulating strain [4,24,25]. The vaccine efficacy (VE) of seasonal influenza vaccines varies and is partially explained by the degree of mismatch seen that year and the choice of vector [25]. For example, during the 2008–2009 flu season, the average influenza vaccine efficacy (VE) was 70% and 38% for inactivated influenza vaccines (IIAV) and the LAIV, respectively, with IIV VE decreasing significantly over time [26]. During the 2015–2016 flu season, LAIV efficacy reached a record low of 3% among 2–17 years old children in the United States [27]. This, along with relatively poor efficacy in the previous two seasons, led the US Centers for Disease Control and Prevention (CDC) to recommend that clinicians not administer LAIVs the following season [27].

The variable VE has prompted calls for new solutions around influenza vaccine design [28,29,30]. We herein describe the theories underlying the design of current influenza vaccines (both universal and seasonal) in the context of immune responses to influenza virus antigens. We additionally present antigenic drift and thrift-based models for universal influenza vaccine design as an alternative to the current prevailing approaches [31,32].

### 1.1. Antigenic Drift

Influenza is generally thought to evolve by the process of antigenic drift. Antigenic drift states that the virus escapes population immunity through the incremental accumulation of mutations in surface proteins [33,34,35,36,37]. Strains containing advantageous escape mutations spread throughout the host population to become the most prevalent seasonal strain (Figure 1). The build-up of escape mutations reduces VE and requires the seasonal vaccine to be updated [36,38,39].

There is a large amount of experimental evidence supporting antigenic drift. Strong evidence for drift comes from (i) antibody escape mutants and (ii) haemagglutinin inhibition (HAI) assays using sera produced by vaccination with historical isolates. For example, Caton et al. demonstrated that the H1 PR8 virus could produce escape mutants when cultured with an array of monoclonal antibodies [40]. More recent studies have since repeated the experiment using more recent strains, such as the 2009 H1N1 pandemic virus [41].

Other studies involved vaccinating naïve ferrets with historical influenza strains. Sera from the vaccinated ferrets was then used in HA inhibition (HAI) assays to determine if it could neutralise historical strains. These studies tended to show that sera raised against a single virus only neutralised chronologically similar viruses [5]. Some studies have gone further and analysed large sets of HAI assay using antigenic cartography, finding that the evolutionary trajectory of influenza occurs in a linear manner, suggesting that antigenic distance increases over time, a key feature of antigenic drift [37].

Whilst influenza is perceived to be a highly variable pathogen due to its fast mutation rate and capacity to generate escape mutants [40,42], epidemiological and phylogenetic studies (Section A.2) have shown that the genetic and antigenic diversity of influenza is substantially restricted [36,37,43]. Influenza seasons in the Northern and Southern hemispheres tend to be dominated by a single or limited number of strains [44,45]. For this reason, seasonal vaccines containing a single H1N1, H3N2, and one or two influenza B lineages often provide non-negligible protection against illness and death. This discrepancy creates a paradox at the centre of the influenza field [46]. The variability of influenza is hard to reconcile with the epidemiological data, where single or limited strain dominance is observed [46]. The paradox can be reduced to the following question: if we perceive influenza being highly variable, then why do we not observe numerous co-circulating lineages of influenza (Figure 1)? In fact, why can we efficiently vaccinate against influenza at all?

Many studies, invoking an array of evolutionary, epidemiological, and mathematical tools, have tried to explain the limited observed diversity of influenza whilst maintaining the antigenic drift theory. Rambaut et al. suggested that a selective sweep (i.e., fixation of a certain strain due to selective pressure) occurs in influenza when strains are exported from source populations into sink populations [45]. They hypothesised that regions where influenza seasonality is weak, such as the tropics, may serve as the reservoir populations for diversity. Subsequently, the diversity is severely curtailed when strains are exported into locations with strong seasonality and hence more severe selective pressure, such as the Northern and Southern hemispheres.

Other studies have employed mathematical models to tackle this problem. Ferguson et al. modelled the evolution and spread of influenza under a variety of conditions [47]. They found that to be compatible with the limited diversity and mutation rates of influenza, a secondary, short-lived, non-strain-specific immunity was necessary. Such an immune response can facilitate competitive exclusion between strains, effectively limiting their diversity. A more parsimonious model of influenza evolution by Tria et al. reached similar conclusions—non-strain-specific immunity is required to bound the diversity of influenza strains under antigenic drift [48].

Koelle et al. suggested the existence of clusters of genotypes providing similar antigenic phenotypes (leading to high cross-immunity) within clusters and dissimilar antigenic phenotypes (leading to low cross-immunity) between clusters [49]. An emergent phenotype, or cluster, will therefore have low population immunity. Hence, one strain can curtail the genetic diversity of previous phenotypes due to its substantial selective advantage over existing phenotypes.

A related approach was undertaken by Bedford et al., who showed that mutations coding for novel antigenic variants can lead to canalisation of influenza evolution [43]. That is, previous exposure of the host population to influenza viruses leads to a fitness landscape (i.e., the mapping from genotypes to fitness that consists of a main trajectory) [50]. Strains on this evolutionary trajectory are likely to outcompete offshoot strains and hence keep diversity low. Notably, various other models have also considered between-strain competition as a factor reducing diversity [51,52,53].

Yuan and Koelle further postulated that restrictions in receptor avidity acting as the dominant selective pressure can act to limit the diversity of influenza strains [54]. Receptor avidity could be a mechanism whereby the fitness of strains limits diversity [28,55]. Similarly, Bush et al. demonstrated that 18 codons under positive selection within the RBS and surrounding antigenic sites could be used to retrospectively predict the emergence of the dominant strain for 9 out of 11 influenza seasons between 1986 to 1997 [56].

Genetic bottlenecks have also been explored as potential processes supporting the observed diversity under antigenic drift [57]. Such bottlenecks can occur in various stages of infections (e.g., during expulsion from a host or inoculation of new hosts), where they limit the population size and can therefore curtail diversity. Analogously, differing intensities of selection in various stages of infection dynamics were observed for influenza and proposed as factors affecting influenza diversity patterns [58,59]. Whereas most influenza models focus on between-host dynamics, a few have explicitly modelled within-host evolution of influenza [59,60,61]. Notably, a recent study by Morris et al. [60] modelled both within- and between-host influenza dynamics and assumed that influenza strains undergo asynchronous selective pressure during transmission and replication. They showed that under such assumptions, the emerging dynamics of influenza strains indeed exhibit constrained antigenic evolution.

Despite all the above-mentioned models and their substantive contributions to our understanding of influenza evolution, there is still no consensus regarding the mechanisms reconciling the paradox underlying the antigenic evolution of influenza. This has important implications for vaccine design—if parts of the virus are limiting the diversity of influenza, then these regions or sites would be ideal vaccine targets. If virus diversity is limited by other mechanisms such as seasonality, then creating an influenza vaccine needs to be approached through other means.

### 1.2. Antigenic Thrift

Antigenic thrift theory was developed to reconcile influenza single or limited strain dominance and phylogenetic information regarding influenza evolution with a testable mechanism [31,44]. Figure 1 highlights the ladder-like, or imbalanced, tree morphology that both H1N1 and H3N2 exhibit. As mentioned in the previous section, this pattern of strain emergence that does not follow the prediction of the antigenic drift model without applying additional evolutionary or ecological assumptions [44].

The antigenic thrift theory states that population immunity is directed against epitopes of limited variability (ELVs) as opposed to highly variable epitopes [31]. The epitopes of limited variability constrain the number of possible immunological variants of the virus. Consequently, as an influenza strain spreads through the population, population immunity is generated against specific conformations of epitopes. Host population immunity then varies due to births and deaths. Consequently, strains containing similar epitopes to those that have circulated in the past reappear once population immunity against them has waned (Figure 2) [31,62].

Hence, the antigenic diversity of the influenza population is constrained by the interplay between the turnover of population immunity and the limited number of variations through which the targeted epitopes can cycle [31,62]. Until recently, experimental evidence for this theory has been limited. However, a number of studies have arisen that support the model. Supporting evidence typically comes from (i) broadly neutralising antibodies isolated from humans and (ii) serological studies constructed using ferrets and mice.

Thompson et al. presented a bioinformatic thrift-based approach to identify less variable epitopes in the head domain of the H1 HA [28]. They collected sera from children aged 7–11 years old during 2006–2007, who had only been exposed to a limited number of strains. The sera were then used to show that epitopes identified in silico mediated immunity to historical strains. Finally, mice were vaccinated with chimeric HA constructs to elicit antibodies specifically targeting the many variants of this epitope of interest. The reactivity seen in the sera from children was recapitulated in mice and demonstrated via pseudotyped virus microneutralisation assays and influenza virus challenge. In addition, alternative versions of the epitope were shown to react to a complementary subset of chronologically distinct influenza strains. This study provides evidence for the existence of epitopes present in the head domain of HA, which appear to cycle through a limited number of conformations.

In Carter et al., ferrets were vaccinated with a single dose of various seasonal H1N1 influenza strains. These strains produced antibodies that cross-reacted with a number of chronologically distinct strains [63]. For example, vaccination with the A/Den/1/1957 virus isolated in 1957 produced an antibody response that was able to cross-react with strains isolated in 1934 and 1999 but not in 1947, 1978, 1991, 2007, and 2009. However, other studies using similar protocols have not identified such patterns of reactivity to historical strains [5]. Carter et al. suggested that there may be epitopes that reappear in chronologically distinct strains. The study further demonstrated that exposure to certain epitopes in sequence displayed by pre-2009 seasonal strains was able to produce a protective antibody response against the novel 2009 strain [63].

More recently, Andrews et al. used B cell screening to identify cross-reactive antibodies targeting conserved regions of the HA that could be frequently isolated from the population. These antibodies typically bound to epitopes located around the RBS and overlapping regions between the monomers of the trimers (the so called ‘lateral’ patch) as well as the stem. Antibodies targeting these regions were found to be broadly reactive against multiple strains [64].

Several antibodies that neutralise chronologically distinct influenza strains have also been described. In 2011, Whittle et al. isolated a monoclonal antibody which bound to the HA receptor binding site and neutralised 30 out of 36 chronologically distinct H1 strains [65]. Similarly, in 2018, Nogales et al. identified an antibody that was able to neutralise historical and modern H1 strains both in microneutralisation assays and via protection in challenge studies in mice [66]. If broadly reactive antibodies similar to those identified in Whittle et al. and Nogales et al. are commonly found in the human population, as suggested by other studies [64], then the sites they bind to could be the epitopes of limited variability outlined in the antigenic thrift theory. Indeed, the antibody described in Whittle et al. does appear to bind to the same location as that identified in Thompson et al. [28,65].

It should be noted that, like any model, the antigenic thrift model predicts a pattern of behaviour based on certain assumptions. It does not specifically explain every aspect of influenza evolution, such as the role of reassortment; but it rather proposes the major factors at play in the antigenic evolution of influenza. To date, the antigenic thrift model has also only been used to describe the antigenic evolution of influenza A and not influenza B.

Furthermore, the drift and thrift theories are not mutually exclusive. A number of proposed mechanisms for the restriction of antigenic drift [60,62] could act as the basis for the epitopes of limited variability outlined in antigenic thrift [34].

### 1.3. Vaccine Approaches

The current seasonal influenza vaccines (i) have variable efficacy, (ii) need to be updated regularly, and (iii) require that the vaccine strains are selected roughly 6 months prior to the influenza season [45]. Several strategies have been employed to develop a universal influenza vaccine to overcome these problems.

## 2. Vaccines Targeting More Conserved Regions of the Influenza Virus

First, we present vaccine development approaches based on the perception that influenza evolves by mutation of highly variable epitopes throughout the HA head domain. Typically, these vaccines try to target conserved regions of the influenza virion, such as the HA stem or internal influenza proteins.

One approach used to induce stem-targeted antibodies, as outlined in Nachbagauer et al., involved designing a vaccine with chimeric HA constructs that contained the head domain of a non-human-circulating avian HA subtype and an H1 stem [61,67]. Exposure to the same stem domain whilst varying avian head domains generated a potent antibody response against the stem in ferrets [61,68,69,70].

A 2020 study by Amitai et al. similarly used a stem-based approach to universal vaccine design [71]. Amitai et al. generated immunodominance heatmaps based on molecular dynamic simulations to identify a conserved, immuno-subdominant site in the stem domain [71]. They tested the results of these simulations by vaccinating mice with a nanoparticle that presented the stem domain of HA. This clearly differed from the method used by Nachbagauer et al., but the goal was the same: to induce antibodies against the stem rather than the immunodominant head [61,67,68,69,70]. Amitai et al. were able to focus the antibody response on the stem domain through a prime and two boost doses. This work has shown promising early results towards the development of a stem-based vaccine.

In 2019, an H1 stem-targeting version of the Nachbagauer et al. vaccine moved into a phase I clinical trial. The trial involved a prime and a boost with H8 head/H1 stem and H5 head/H1 stem chimeric vaccine constructs as either an IIAV or LAIV. The trial took place in individuals aged 18–39 and the time between prime and boost was 85 days. Interim results showed that the vaccine was unable to produce significant levels of H1 HA stem-targeting antibodies without the addition of the GlaxoSmithKline proprietary adjuvant, AS03 [67]. With the AS03 adjuvant, the IIAV vaccine but not the LAIV vaccine produced significant amounts of H1 stem-targeted antibodies after a single dose. As a result, in 2019 GSK halted further development of the vaccine candidate [72].

In 2020, once the trial was completed, a follow-up paper showed that sera taken from selected trial participants given either the LAIV or IIIV vaccine were able to significantly protect mice via passive transfer from H6N5 influenza virus challenge. This demonstrated that the vaccine could potentially provide some degree of pandemic protection. The approach has been shown to be a safe and feasible vaccination strategy in humans if the vaccine immunogenicity issues can be overcome.

The studies mentioned above assumed that the head domain is too variable to become the basis of a universal vaccine [61,67,68,69,70,71,72]. Hence, these approaches used complex systems that try to modify the tendency of the human immune system to target the immunodominant HA head.

Another method to address the problem of a highly variable head domain has been to target internal viral antigens that can induce CD8+ T cell responses. Influenza challenge studies using the 2009 H1N1 pandemic (H1N1pdm09) and 2013 A/H7N9 outbreak strains have demonstrated that pre-existing antigen-specific CD8+ T cells lowered viral shedding, reduced symptoms [73,74], and led to faster recovery from severe infection [75]. Similar approaches have also shown that pre-existing CD4+ T cells correlate with protection in humans [76].

One such approach uses Chimpanzee Adenovirus (ChAdOx) or Modified Vaccinia Ankara (MVA) viral vectors containing the internal influenza proteins, nucleoprotein (NP), and matrix 1 (M1) [77]. These viral vectored vaccines were shown to have moderate efficacy in a small human challenge study [78] and promising immunogenicity in younger and older adults [79]. However, a phase 2 trial undertaken by Vaccitech using the MVA-based version of the vaccine did not reach its clinical objectives [80].

A further promising approach to influenza vaccine development involves the use of a pseudotype replication-incompetent influenza vaccine (SFLU). This vector system is capable of going through one round of replication, thus presenting internal antigens to the immune system [81]. SFLU has shown promising T cell immunogenicity in mice [81] and pigs [82]. SFLU also reduced viral load and transmission following heterotypic challenge in ferrets, whereas in pigs the vaccine was only capable of reducing pathology [83]. Consequently, the SFLU vaccine has shown some promising immunogenicity data, although is exhibits varying efficacy levels between small and large animal models [83]. However, no T cell influenza vaccine has shown efficacy in large-scale human trials or influenza challenge studies.

There have also been several attempts at developing vaccines targeting neuraminidase (NA), which has proved difficult due to the immunodominance of HA [84,85]. The work of the Krammer Group has advocated for the importance and standardisation of neuraminidase antigens in protective antibody responses against influenza [86]. In a 2020 study, they found that protective antibodies could be induced in mice against enzymatically inactivate NA [84].

Doyle et al. identified a neuraminidase epitope conserved in all influenza A viruses [87]. They were able to show that this site is accessible with monoclonal antibodies and saw success in mice H1N1 and H3N2 challenge studies. Since rabbit monoclonal antibodies against the epitope in question were injected into naïve mice, the difficulty of promoting this site in humans is as yet unknown. It is unclear whether vaccination promoting this epitope would be useful, as most patients would likely already have pre-existing immunity for HAs, making it difficult to subvert the tendency of the immune system to target the dominant HA. This approach was tested in further mouse studies in 2019 but has not begun clinical trials [88].

Indeed, neuraminidase may be a key component in broadly protective vaccines against influenza in the future. These studies suggest that, at the least, the use of NA in vaccines should be more uniform and involve consensus sequences to help augment the immune response elicited by HA.

Another common IIAV universal vaccine target is the tetrameric type 3 transmembrane matrix 2 (M2) protein. M2 is present in small quantities on the virion [89]. M2 has a highly conserved ectodomain domain, M2e, which makes it a candidate for a universal vaccine target [90,91]. M2e antibodies are non-neutralising and are thought to mediate protection via antibody-dependent cell cytotoxicity (ADCC) [92]. Due to M2e’s highly conserved nature, the applicability of the antigenic drift or thrift models to M2e vaccine design is limited, and thus M2e is unlikely to be a major determinant of the antigenic evolution of influenza. Despite this, M2e vaccines have shown promising results via in vitro and animal studies [93,94].

Multiple M2e-targeted vaccines are currently undergoing or have completed a phase I clinical trial. VaxInnate’s VAX102 vaccine consists of four tandem copies of M2e fused to the TLR5 ligand flagellin [95]. VAX102 underwent phase 1 and phase clinical trials but development has since been discontinued [96,97]. Another M2e-targeted vaccine moving through a phase I trial is Uniflu’s HBc/4M2e construct, which comprises M2e proteins fused to the immunodominant loop of the Hepatitis B core (HBc) antigen [98]. ACAM-FLU-A also consists of M2e fused to an HBc. The phase 1 clinical trial reported sera antibody development in 90% of participants [99].

## 3. An Antigenic Thrift Approach to Universal Influenza Vaccine Design

The antigenic thrift theory offers an alternative pathway to the development of a universal influenza vaccine. One corollary of the theory is that vaccination against all typical circulating human strains may be achievable by targeting all the possible conformations of a given set of epitopes. Several approaches are currently targeting epitopes of limited variability or relatively conserved epitopes in the head domain of HA. These either implicitly mention antigenic thrift or are relying on epitopes that the theory suggests should have a broad protective coverage.

This basis for the limited variability of these epitopes could be dictated by the same mechanisms as those proposed in Bush et al. and/or Yuan and Koelle—namely, restriction mutation at a small number of sites. Consequently, the basis for the defined variants’ antigenic thrift could be derived from random mutations occurring via drift. Thus it is important to note that antigenic drift and thrift are not mutually exclusive.

As previously mentioned, Thompson et al. identified a series of epitope variants using bioinformatics and serology in direct reference to the antigenic thrift model [28]. By targeting all possible variants of the epitopes of limited variability, it is expected that a vaccine based on these epitopes should protect against entire subtypes. This approach involves a similar method to Nachbagauer et al., whereby epitopes from the H1 or H3 HA are substituted into avian HA proteins. This approach utilises a structural bioinformatic pipeline to increase the speed and accuracy of vaccine design. Vaccination using a prime-boost regimen then focuses the immune responses against the substituted epitope, shown in mice in Thompson et al. This vaccine design includes considerations of epitope variability, species lifespan, and cross-reactivity based on the mathematical model described by Recker et al. and Wikramaratna et al. [31,32]. In 2019 this approach was licensed to the start-up company, Blue Water Vaccines, and a vaccine following this method is currently under development [100,101].

Other approaches are also identifying broadly reactive epitopes in the head domain of HA for use as vaccine targets. Utilising the findings in Carter et al., the Ross group has developed a vaccine displaying computationally optimised antigens (COBRA) for the H1, H3, and H5 HAs [102,103,104]. The optimised antigens were each designed to be representative of certain timepoints. For example, the X2 HA COBRA was developed based on H1 HA sequences from 1933 to 1947 [102]. This antigen is therefore representative of a period shortly after the first recorded influenza pandemic, early in the evolutionary history of influenza in humans. The COBRA HAs were first tested in mice using a virus-like particle delivery method in a mixture, alone, and in prime-boost combinations. These candidates were then challenged with historical and circulating strains. A mixture of four COBRA VLPs delivered in a specific prime-boost format was found to be most successful for H1 [85], with similar results for H5 [104], whilst the COBRA antigens showed broad neutralisation on their own for H3 [103].

The COBRA vaccine has now been developed into an inactivated split vaccine presenting both H1 and H3 antigens, which has thus far been tested on ferrets. The authors of this study note that the antibody titres produced in the ferrets’ post-vaccination, against both H3N2 and H1N1 in HA inhibition assays, were reduced compared to studies using virus-like particle vaccines [29,105]. Although this vaccine candidate showed only mild success, it is a promising start towards the HA head-focused broadly protective vaccine.

Finally, DIOSynVax, (Digitally designed, Immune Optimised Selected and Synthetic Vaccines) are using bioinformatics tools to identify conserved immunogenic regions within HA. Specifically, DIOSynVax draws on sequence information for both influenza and known immune correlates (i.e., antibodies) to design vaccine candidates. These designed antigens are then expressed via DNA vaccine vector. Once expressed via DNA vectors, candidates are screened for broad reactivity and effectiveness. So far, this vaccine remains in pre-clinical development. However, DIOSynVax advocates for the pre-clinical success of this approach through as yet unpublished work on an Ebola, Marburg, and Lassa fever vaccines [106].

It is likely that many of the regions identified by DIOSynVax could be classed as epitopes of limited variability. Indeed, all of these approaches share a common feature in that they identify conserved regions of the major influenza antigen HA, which produce broadly reactive responses upon vaccination against chronologically distinct strains. These targets are predicted to exist by some interpretations of drift [54,56] or the antigenic thrift model. The success of these three approaches reinforces the existence of regions of limited variability within the influenza virus that can be exploited to produce a universal influenza vaccine.

## 4. Pitfalls of Effective Universal Vaccine Development

With the development of vaccines with greater efficacy for H1N1 and H3N2 comes a heightened risk for immune escape if full immunity to all possible influenza variants is not conferred. Increased VE against entire subtypes could remove the H1N1 and H3N2 strains from circulation, thus likely creating the right conditions for a significant zoonotic transfer leading to a pandemic. Zhang et al. discussed this notion in the wake of the 2009 H1N1 pandemic [107,108]. This threat underscores the importance of working towards a universal vaccine, from which immune evasion would be less likely and also highlights that caution needs to be applied when deploying such vaccines [109].

## 5. Concluding Remarks and Future Perspectives

Many universities, start-ups, and pharmaceutical companies are trying to develop universal influenza vaccines. The focus of many of these approaches is to avoid variable parts of the influenza virus. The rationale for these approaches is that targeting more conserved parts may allow a single vaccine to protect against a large number of current and future influenza strains. Although these approaches have produced promising results in many instances, so far the goal of a universal influenza vaccine has not been achieved. However, at the heart of how we view the evolution of influenza is a paradox: we perceive influenza as being highly variable, yet only a limited number of strains circulate each season. Exploring the evolutionary theory that underpins our vaccination strategies could therefore lead to improved influenza vaccines in the future.

## Figures and Tables

**Figure 1 vaccines-09-00657-f001:**
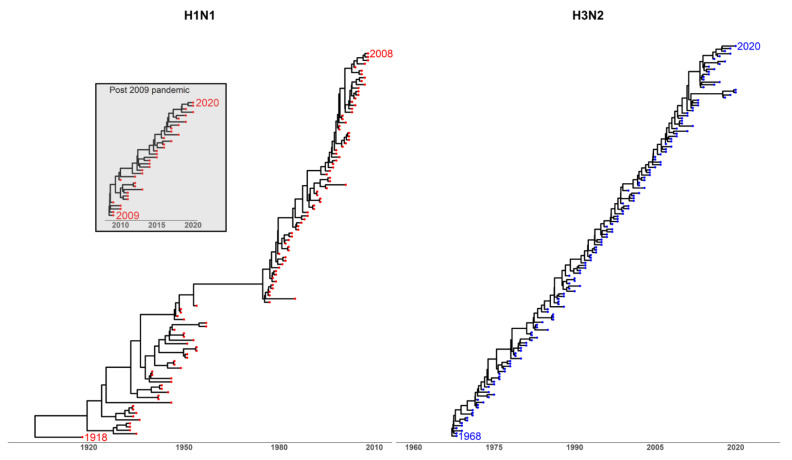
Maximum clade credibility trees of human-circulating influenza A viruses (IAV) HAs. H1N1 strains sampled between the years 1918 and 2020. H3N2 strains sampled between 1968 and 2020. Three human-circulating strains per sampling date were, where possible, randomly sampled for each subtype (Section A.1).

**Figure 2 vaccines-09-00657-f002:**
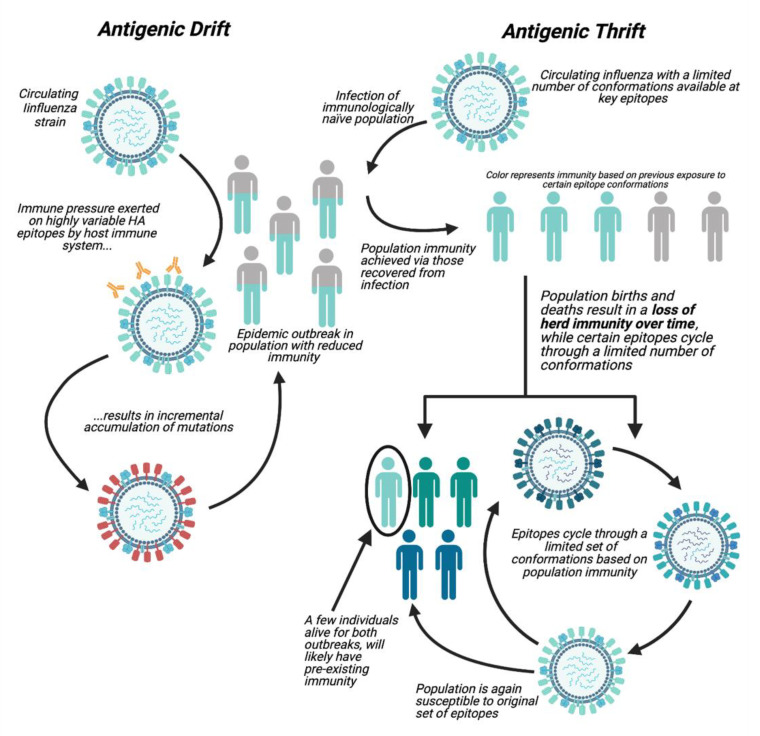
A comparison of the antigenic drift and thrift theories. Antigenic drift theory states that influenza is highly variable and escapes population immunity through the accumulation of incremental mutations over time. Conversely, the antigenic thrift theory states that population immunity is directed against epitopes of limited variability (ELVs) [31,44]. Population immunity to these ELVs changes over time due to births and deaths in a population. This allows for the reappearance of historical strains once immunity against them has waned.

## Data Availability

HA data were obtained from the Influenza Research Database (IRD). “Nucleotide Sequence Search” (see https://www.fludb.org/brc/influenza_sequence_search_segment_display.spg?method=ShowCleanSearch&decorator=influenza, accessed on 16 March 2021) was used to obtain human HA sequences [110]. As of 16 March 2021, 23,418 and 28,234 sequences of H1N1 and H3N2, respectively, were available.

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
