# Peer review of "An Antigenic Thrift-Based Approach to Influenza Vaccine Design"

_vaccines, 2021, doi:10.3390/vaccines9060657_

Round 1

Reviewer 1 Report

The authors reviewed influenza vaccine papers to develop universal flu vaccines. Overall, the authors have made a good attempt at adding value to the discussion of ideas to develop universal flu vaccines. However, a minor revision of manuscript is needed before it can be accepted for publication.

  1. Can the authors discuss about antigenic shift and the role of anti-NA antibodies?
  2. Line 259: The sentence doesn’t flow well. “In 2019, this caused GSK to drop further development” What does “this” mean?
  3. Please decipher abbreviations after the first incidence (Line 88 & 94; HAI)
  4. Please explain the meaning of horizontal bar. For example, what happened for H1N1 between 1955 and 1970. Nothing happened? Why disconnected after 2009? Can you explain by antigenic thrift theory?
  5. Line 257: “ASO3” should be “AS03”.

Author Response

We would like to thank reviewer 1 for their helpful comments, which we have tried to address below.

  1. Can the authors discuss about antigenic shift and the role of anti-NA antibodies?

We have added an overview of antigenic shift between lines 80-95 and a summary of NA antibodies and vaccines between lines 585-605.

  1. Line 259: The sentence doesn’t flow well. “In 2019, this caused GSK to drop further development” What does “this” mean?

We have clarified this statement and it now reads: “As a result, in 2019 GSK halted further development of the vaccine candidate [75]”.

  1. Please decipher abbreviations after the first incidence (Line 88 & 94; HAI)

We have done this. Thank you for pointing it out.

  1. Please explain the meaning of horizontal bar. For example, what happened for H1N1 between 1955 and 1970. Nothing happened? Why disconnected after 2009? Can you explain by antigenic thrift theory?

This is an intriguing aspect regarding the history of influenza. H1N1 influenza emerged in 1918, causing the Spanish flu. It then became endemic, circulating until 1957 when it was replaced by H2N2. Incredibly, in 1977, the H1N1 strain that circulated in 1948 was released and circulated until 2009. It’s thought the release was from either a Chinese or Russian laboratory (1).

The reappearance of a strain 1977 that stopped circulating in 1957 demonstrates that in the 20 years hiatus, population immunity had sufficiently changed to enable the strain to continue circulation. Thus in some ways this would support the concept of antigenic thrift as this theory states that population immunity to epitopes of limited variability dictates the circulation or not of a given strain. In 1957 population immunity didn’t allow the H1N1 strain to circulate further, but in 1977, due to births and deaths within the population changing population immunity, the H1N1 strain could continue to circulate.

References: 

  1. Kendal, A. P., Noble, G. R., Skehel, J. J. & Dowdle, W. R. Antigenic similarity of influenza A(H1N1) viruses from epidemics in 1977-1978 to ‘Scandinavian’ strains isolated in epidemics of 1950-1951. Virology89, 632–636 (1978).

Reviewer 2 Report

This manuscript is an on-going elaboration and advocacy for the “antigenic thrift” model on influenza virus evolution. Current paradigm is that influenza virus evolves through antigenic drift, and for influenza A virus, antigenic shift as well. Antigenic drift is by random mutations with nonsynonymous substitutions, particularly at antigenic sites, and selected by immune pressure. It is welcoming to have alternative theories to reconcile any inconsistence from current available data. It may potentially result in a “paradigm shift” for this field. However, this manuscript lacks a vigorous discussion using comprehensive data, related mechanisms, and other theories in the context of this thrift theory. Any incompatibilities and inconsistence regarding this thrift theory should also be identified and discussed.

The following major points should be addressed prior to publication:

  1. Despite the "randomness" of mutations, evolution of influenza A virus has been shown to utilize mechanisms such as co-circulation, reassortment, and alternate circulation (too many references to be listed here). As mentioned above, the omission of discussion with reference to these mechanisms is disappointing. A revised manuscript should include the applicability and compatibility of the thrift-based model to explain the above-mentioned evolutionary mechanisms 
  2. Bush et al. had shown that there were disproportional nonsynonymous substitutions at the antigenic sites. Furthermore, antigenic cartographic studies had shown that antigenic evolution for influenza A virus was stepwise--as antigenic clusters--rather than continuously. These results do not conflict with the antigenic drift model. Again, it is pertinent that the thrift-based model is compatible to these observations
  3. By authors' own admission, if the average length of an epitope is 15 amino acids, and there are 5 antigenic sites (for H3), the possible combination will be 5 X 20^15--an astronomical large number. The number of available variants, whether it is recycled by the thrift theory, or random drift by the drift theory, doesn't make any difference.  Therefore, it is more of semantics than practical. Likewise, the evolution of "variants" still depends on the selective pressure in both cases
  4. Despite the catchy title: “new approach to influenza vaccine design”, there was only a very brief description on current research--only 2.5 pages long--without any detailed discussion or description on the mechanisms and their relevance to support the thrift-based model

Minor points:

  1. Fig. 1 is misleading, as only one strain was sampled per data point. There have been many published phylogenetic trees showing co-circulation of more than one strains for up to 2 yr before the “dominant” strain appeared
  2. Reference #28 is missing the journal (PNAS?!)
  3. A brief elaboration on the applicability of this thrift model on influenza B virus should also be provided, for completeness

Round 2

Reviewer 2 Report

This revised version has been significantly improved, as most of the concerns raised have been addressed. Instead of changing the title, the authors should insert a statement that their model applies to influenza A virus, not influenza B virus, and preferably after the paragraph that ends at Line 253.

Author Response

We would like to thank the Reviewer for taking the time to read the manuscript and for their assistance in greatly improving it.

We have added the sentence at line 256: “To date, the antigenic thrift model has also only been used to describe the antigenic evolution of influenza A and not influenza B".